# Therapeutic Targeting of Protein Disulfide Isomerase PDIA1 in Multiple Myeloma

**DOI:** 10.3390/cancers13112649

**Published:** 2021-05-28

**Authors:** Metis Hasipek, Dale Grabowski, Yihong Guan, Raghunandan Reddy Alugubelli, Anand D. Tiwari, Xiaorong Gu, Gabriel A. DeAvila, Ariosto S. Silva, Mark B. Meads, Yvonne Parker, Daniel J. Lindner, Yogen Saunthararajah, Kenneth H. Shain, Jaroslaw P. Maciejewski, Frederic J. Reu, James G. Phillips, Babal K. Jha

**Affiliations:** 1Department of Translational Hematology and Oncology Research, Cleveland Clinic Foundation, Taussig Cancer Institute, Cleveland, OH 44195, USA; hasipem@ccf.org (M.H.); grabowd@ccf.org (D.G.); guany2@ccf.org (Y.G.); TIWARIA@ccf.org (A.D.T.); GUX@ccf.org (X.G.); PARKERY2@ccf.org (Y.P.); lindned@ccf.org (D.J.L.); saunthy@ccf.org (Y.S.); maciejj@ccf.org (J.P.M.); phillij5@ccf.org (J.G.P.); 2Department of Malignant Hematology, H. Lee Moffitt Cancer Center and Research Institute, Tampa, FL 33612, USA; RaghunandanReddy.Alugubelli@moffitt.org (R.R.A.); gabriel.deAvila@moffitt.org (G.A.D.); Ken.Shain@moffitt.org (K.H.S.); Frederic.Reu@moffitt.org (F.J.R.); 3Department of Cancer Physiology, H. Lee Moffitt Cancer Center and Research Institute, Tampa, FL 33612, USA; Ariosto.Silva@moffitt.org (A.S.S.); Mark.Meads@moffitt.org (M.B.M.); 4Lerner College of Medicine, Cleveland Clinic Foundation, Cleveland, OH 44195, USA; 5Case Comprehensive Cancer Center, Case Western Reserve University, Cleveland, OH 44106, USA

**Keywords:** protein disulfide isomerase PDIA1, ER stress, IRMM, ERMM, UPR

## Abstract

**Simple Summary:**

Multiple myeloma (MM) is a cancer of antibody-producing plasma cells that remains incurable. These cells heavily depend on protein disulfide isomerase, PDIA1, for folding and structural integrity of antibodies and other secretory proteins to avoid unresolvable stress caused if they remain unfolded. High PDIA1 expression confers resistance to proteasome inhibitors and other stressors due to the gain in endoplasmic reticulum (ER) function, while maintaining or increasing vulnerability to PDIA1 inhibition. Here we report the identification and characterization of an orally bioavailable novel PDIA1 inhibitor CCF642-34 that is effective against multiple myeloma in pre-clinical models. PDIA1, the ER resident enzyme essential for the folding of disulfide bond-containing proteins, is upregulated in relapse and refractory myeloma. This increase in PDIA1 confers its sensitivity to CCF642-34, a structurally optimized PDIA1 inhibitor that induces apoptosis in myeloma cells but not in normal bone-marrow-derived CD34+ hematopoietic stem and progenitor cells.

**Abstract:**

Multiple myeloma is a genetically complex hematologic neoplasia in which malignant plasma cells constantly operate at the maximum limit of their unfolded protein response (UPR) due to a high secretory burden of immunoglobulins and cytokines. The endoplasmic reticulum (ER) resident protein disulfide isomerase, PDIA1 is indispensable for maintaining structural integrity of cysteine-rich antibodies and cytokines that require accurate intramolecular disulfide bond arrangement. PDIA1 expression analysis from RNA-seq of multiple myeloma patients demonstrated an inverse relationship with survival in relapsed or refractory disease, supporting its critical role in myeloma persistence. Using a structure-guided medicinal chemistry approach, we developed a potent, orally bioavailable small molecule PDIA1 inhibitor CCF642-34. The inhibition of PDIA1 overwhelms the UPR in myeloma cells, resulting in their apoptotic cell death at doses that do not affect the normal CD34^+^ hematopoietic stem and progenitor cells. Bortezomib resistance leads to increased PDIA1 expression and thus CCF642-34 sensitivity, suggesting that proteasome inhibitor resistance leads to PDIA1 dependence for proteostasis and survival. CCF642-34 induces acute unresolvable UPR in myeloma cells, and oral treatment increased survival of mice in the syngeneic 5TGM1 model of myeloma. Results support development of CCF642-34 to selectively target the plasma cell program and overcome the treatment-refractory state in myeloma.

## 1. Introduction

Multiple myeloma (MM) is a genetically complex hematological malignancy which is characterized by clonal proliferation of plasma cells in the bone marrow and secretion of monoclonal antibodies and cytokines that can damage bone, bone marrow, and kidney function [1]. Although clinical outcome continues to improve with introduction and investigation of novel agents, the presence of genetically heterogeneous sub-clones essentially precludes cure [2]. Per SEER (https://surveillance.cancer.gov/joinpoint/, accessed on 25 March 2021) estimates, MM was the cause of death for 12,830 individuals in the US in 2020, while 32,270 were newly diagnosed.

MM cells carry the highest protein synthesis and secretory burden of all mammalian cells [3], amplifying the high dependence on the unfolded protein response in cancer [4] to a degree where proteasome inhibition provided a major breakthrough [5]. Despite the success of proteasome inhibitors and other recently approved drugs, including CD38-targeting antibodies, MM remains incurable in most patients. Importantly, the refractory state to current drugs portends poor median survival below 6 months [6], indicating a persistent unmet medical need. The secretion of large quantities of immunoglobulin (Ig) and cytokines by myeloma cells requires re-arrangement of intramolecular disulfide bonds after their translation from mRNA [3,7,8]. Protein disulfide isomerases (PDIs) are the only enzymes to meet this need through their reductase, oxidase, and isomerase functions [9,10], and PDIA1 is the main endoplasmic reticulum (ER) resident isoform of this multifunctional protein family [11,12]. PDIA1 is upregulated in multiple malignancies such as melanoma, lymphoma, hepatocellular carcinoma, brain, kidney, ovarian, prostate, and lung cancers [13,14,15,16,17]. The ER-based functions of PDIA1 as integral parts of the unfolded protein response have been linked to the “Achilles heel” of MM [18]. High protein synthesis, nutrient deficiency, and hypoxia in MM cause the ER to function at maximum capacity [4] where perturbation results in cell death. To date, this has only been exploited clinically through proteasome inhibition, suggesting that targeting additional adaptive responses may help counteract the proteasome inhibitor refractory state and provide new myeloma selective treatment options.

Previously, we reported the identification of a PDI inhibitor (CCF642) from a phenotypic multilayered MM cell-based cytotoxicity assay that modeled disease niche, normal liver, kidney, and bone marrow [19]. CCF642 covalently modified the catalytic site lysine residue leading to PDIA1 inactivation, inducing irreversible lethal ER stress and hence elimination of MM cells both in vitro and in vivo with no apparent adverse effects on normal bone marrow cells [19]. In addition, CCF642 maintains its therapeutic effect against bortezomib (BTZ)-resistant MM cells through PDIA1 inhibition. However, CCF642 has poor solubility and suboptimal selectivity precluding clinical translation [19]. Here, we show that expression of PDIA1 inversely correlates with survival in relapsed and refractory myeloma patients, and using structure-guided medicinal chemistry, we developed a new analogue of CCF642. The new PDIA1 inhibitor, CCF642-34, specifically binds and inhibits PDIA1. Unlike CCF642, it has improved drug-like properties, including improved solubility, selectivity, and potency, and is effective when administered orally in an aggressive syngeneic mouse model of myeloma.

## 2. Materials and Methods

Cells and reagents: MM1.S-luc-BTZ (BTZ^®^, Wilmington, NC, USA)-resistant cell line was made in our laboratory. Starting with 1 nM concentration, MM1.S-luc cells were treated with BTZ and exposed continuously for 5 days and removed for 2 days before re-exposure to the drug until the growth of the cells mimicked the parental cell line. Incremental increase of the drug was applied until the concentration of 8 nM was reached. Cell lines were grown according to the guidelines by the supplier and used within 10 passages in fresh culture. di-E-GSSG was from IMCO Corp. Ltd. IMDM and RPMI-1640 cell culture media were from Cleveland Clinic media core services. BTZ was procured from Millennium Pharmaceuticals Inc. All cell lines that are used in this study and their detailed information are shown in Table 1 below.

Cell viability assay: Cell viability was measured in 96 well culture plates (2 × 10^4^ cells/well) using CellTiter-Glo^®^ Luminescent Cell Viability Assay (Promega, Madison, WI, USA) according to the manufacturer’s protocol.

In vitro colony-forming assays: Mononuclear cells derived from bone marrow or purified CD34^+^ cells from a healthy donor, cord blood, or MM cell RPMI-8226 were grown in semi-solid methylcellulose media (MethoCult™, H4435; STEMCELL Technologies, Vancouver, BC, Canada) in the presence of indicated compounds and concentrations. A total of 10,000/mL normal bone marrow CD34^+^ cells and 1000/mL RPMI-8226 cells were plated, and colonies were scored on day 14.

Immunoblotting: Immunoblotting was performed as described previously [19,20,21] using primary antibodies against PDIA1 (Cat #3501), XBP1-S (Cat #12782); IRE1α (Cat #3294), C/EBP homologous protein (CHOP) (Cat #5554), Caspase-3 (Cat #9665), PARP1 (Cat #9542), and GAPDH (Cat # 3683) purchased from Cell Signaling Technology, Inc. (Danvers, MA, USA) and used at 1:1000 dilution, unless mentioned otherwise.

Mass spectrometry: Tryptic peptide mixtures were analyzed by online LC-coupled tandem mass spectrometry (LCMS/MS) on an Orbitrap mass spectrometer (Thermo Fisher) as described previously [22,23]. The Sequest software was used to perform database searches, using the Extract_msn.exe macro provided with Xcalibur (version 2.0 SR2; Thermo Fisher Scientific, Waltham, MA, USA) to generate peak lists. The following parameters were set for creation of the peak lists: parent ions in the mass range 400–4500, no grouping of MS/MS scans, and threshold at 1000. A peak list was created for each analyzed fraction (i.e., gel slice) and individual Sequest searches were performed for each fraction. The data were searched against *Homo sapiens* entries in the Uniprot protein database. Carbamidomethylation of cysteines was set as a fixed modification, and oxidation of methionine was set as a variable modification. Specificity of trypsin digestion was set for cleavage after lysine or arginine, and two missed trypsin cleavage sites were allowed. The mass tolerances in MS and MS/MS were set to 10 ppm and 0.6 Da, respectively, and the instrument setting was specified as “ESI-Trap.”

Medicinal chemistry: All of the new PDI inhibitors reported in Appendix A and represented by the generic structure in results were prepared in similar fashion following the chemistry scheme shown for 642-34 in Appendix A and described in detail earlier [19]. The 7 step preparation of the HCl salt of primary amine intermediate 8, which follows standard literature described procedures, begins with synthesis of the benzaldehyde-derived Schiff base 2 of commercially available 4-amino phenol 1 via reflux in excess trimethyl orthoformate. Alkylation of the phenol of 2 with 3-(BOC-amino) propyl bromide in dry DMF and 3 equivalents of cesium carbonate at 50 °C for 15 h gave 3. Catalytic reduction and hydrogenolysis with Pd(OH)_2_ and ammonium formate in refluxing ethanol provided aromatic amine 4. Treatment of 4 with thiophosgene in 1:1 dichloromethane/water gave the crude isothiocyanate 5, which upon reaction with methyl thioglycolate in dichloromethane in the presence of triethylamine afforded the thiazolidinone 6. Condensation of the thiazolidinone 6 with 5-nitro-thiophene-2-carboxaldehyde in acetic acid in the presence of 5 equivalents of sodium acetate at 90 °C for 16 h provided the substituted N-BOC-protected rhodanine 7 upon cooling and precipitation with the addition of water. N-BOC deprotection with excess 4N HCl in dioxane at room temperature (20 °C) for 2 h provided the key intermediate HCl salt 8, which was used to make the 7 analogues shown in Appendix A. The coupling of 8 with the requisite L-amino acids was accomplished in dry DMF with Hexafluorophosphate Azabenzotriazole Tetramethyl Uronium, (HATU, 1-[Bis(dimethylamino)methylene]-1*H*-1,2,3-triazolo[4,5-*b*]pyridinium 3-oxid hexafluorophosphate) and 2 equivalents of Hunig’s base and gave the penultimate intermediates. The final compounds CCF642-34 through CCF642-41 were prepared by treatment with excess 4N HCl in dioxane at room temperature. The biotin derivative CCF642-34-biotin was synthesized from the HCl salt of CCF642-34 by treatment with biotin in DMF with HATU and Hunig’s base. All final compounds showed ^1^H, ^13^C NMR, and Mass Spec analyses consistent with assigned structure. See spectral data included in the Appendix A for CCF642-34, -34 Biotin, -35, 38, and -40.

Computational modeling and molecular dynamics: The computational model used the reduced PDI (NP_000909.2) crystal structure [24] for binding of CCF642 and biotinylated CCF642-34 to PDIA1. Initial docking of CCF642 and its analogues on PDI (PDB ID: 4EKZ) was performed by using AutoDock 4.1 (in AutoDock Tools 1.5.2) as described previously [19]. Binding energy calculations used CHARMM force field in Discovery Studio 1.3 pipeline (Accelrys, Inc., San Diego, CA, USA). The predicted structure of PDIA1-CCF642 covalent complex was used for the generation of structure–activity relationships that were further probed with in vitro enzyme activity assays.

PDI activity: All PDI activity assays were performed using two different substrates, insulin and di-eosin-diglutathione (di-E-GSSG) as described earlier [19,21] with brief modifications. PDIA1 (1 µM) was incubated for 1 h with varying concentrations of PDIA1 inhibitors 642 or 642-34 (0.1, 03, 1, 5, 10, or 20 µM) in 100 mM sodium phosphate pH 7.0, 2 mM EDTA, and 1% DMSO. Bovine insulin (100 µM) and DTT (1 mM) were added to initiate the reaction. Kinetic readings were taken every minute for 2 h at 650 nm absorbance using a BioTek Synergy plate reader (BioTek Instruments, Inc., Winooski, VT, USA). For highly sensitive fluorescence-based assays, di-E-GSSG as a pseudo substrate was utilized to access the activity of PDIA1 [25]. Known concentrations of recombinant PDIA1 were mixed with 10 mM GSH and incubated at 37 °C for 30 min in 150 mM K_2_HPO_4_/KH_2_PO_4_ (pH 7.1) buffer solution. PDI inhibitor was added into the mixture and incubated for an additional 30 min at 37 °C. Di-E-GSSG was added into the reaction mix at a final concentration of 100 nM and samples were transferred into white multi-well plates (Becton Dickson Labware, Franklin Lakes, USA). Synergy H1 plate reader (BioTek, WI, USA) was used for kinetic analysis using excitation at 518 nm, emission at 545 nm, reads of 0.1 s/well every minute at 25 °C for 1 h. Baseline fluorescence was determined from di-E-GSSG reactions without PDI and GSH.

Enzyme kinetics and data analysis: All kinetic analysis was performed using irreversible inhibition methods described earlier [26,27]. K_i_, the inactivation constant, and k_2_, the rate of inactivation, were calculated according to PDI activity at different drug concentrations defined as total occupancy of the active site at exp(k_obs_*time) and the k_obs_ = k_2_[I]/(K_i_ + [I]) for the reaction assuming the following equation:

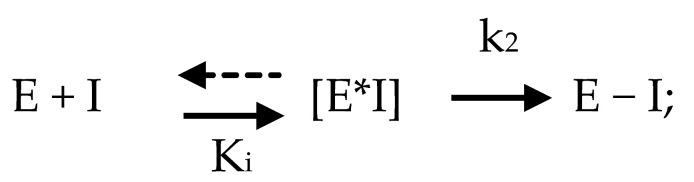

where E is the enzyme, I is the PDIA1 inhibitor, K_i_ is the inactivation constant, and k_2_ the rate of inactivation. All curve fitting and analyses were performed in GraphPad Prism.

PDIA1 purification: Recombinant human PDIA1 (Acc# P07237) was cloned into expression vector pET6xHN-N (Clontech Laboratories, Inc., Ann Arbor, MI, USA) by using Sal1 and Not1 restriction enzymes after amplification with primers that contained Sal1 and Not1 digestion sites (forward: CTCTGTCGACCTGCGCCGCGCTCTG, reverse: AGCGGCCGCTTACAGTTCATCTTTCACAGCTTTCTG). The pET6XHN-PDIA1 was expressed using the *Escherichia coli* strain BL21 (DE3) NiCo (NEB cat# C2529H). This plasmid encodes a fusion protein containing the entire human PDI sequence with an N-terminal His_6_ tag. Recombinant PDI was purified from the soluble fraction of the cell lysate using His60 Ni Superflow resin (Clontech Laboratories, Inc.). Bound PDI was eluted according to the user’s manual (Clontech Laboratories, Inc.) Protein quantification was performed by the Bradford assay.

Liver microsome assay: 20 µM CCF642 or CCF642-34 were incubated with 1 mM NADPH and 0.25 mg/mL human liver microsomes (Sigma, Cat. No: M0567) at 37 °C. Human liver microsomes were precipitated by quenching the reaction into ice-cold acetonitrile at indicated time points. Supernatant was obtained after centrifugation at 15,000× *g* for 5 min and the remaining compound in supernatant was analyzed by Agilent 1260 Infinity II HPLC with Ultra-violet (UV) detector using Gemini column, 3 µM particle size 150 mm × 2 mm (Phenomenex). A gradient of 50/50 acetonitrile/water with 0.1% (*v*/*v*) formic acid was run isocratically for 2 min at 0.3 mL/min flow while maintaining 55 °C column temperature. A gradient of the 90/10 acetonitrile/water with 0.1% (*v*/*v*) formic acid at 0.3 mL/min flow was introduced from 3 min to 13 min. A gradient of 50/50 acetonitrile/water with 0.1% (*v*/*v*) formic acid then ran isocratically for 2 min with the same flow rate, followed by a gradient increase to 100% acetonitrile over 2 min to store the column. The relative amount of drug at each time point was determined by using the UV peak detection at wavelength 254 and 282 nm. This was compared with the control run where human liver microsomes were not present in order to determine remaining drug percentage. Each HPLC-UV run was performed in duplicate.

ROS detection: MM1.S, MM1.S.LUC, and BTZ-resistant MM1.S.LUC cells were washed with DPBS and treated with 5 µM carboxy-H_2_DCFDA (Molecular Probes) for 45 min at 37 °C. The cells were then washed 3 times in DPBS and followed by 2.5 µM CCF642-34 treatment up to 4 h. After the incubation, the cells were washed 3 times and the intracellular ROS was quantified (excitation = 493 nm; excitation = 523 nm) using a BioTek Synergy plate reader (BioTek Instruments, Inc., Winooski, VT, USA).

RNA-seq and analysis: MM1.S cells were treated with 3 µM of either CCF642 or CCF642-34 for 6 h, and RNA was purified by using the NucleoSpin RNA kit (Takara Bio USA, Inc.; cat. #740955, Mountain View, CA, USA) according to the manufacturer’s instruction. The RNA sequencings were completed as reported previously [23]. The RNA-seq data were submitted to the Gene Expression Omnibus (GEO) repository at the National Center for Biotechnology Information (NCBI) archives, with assigned GEO accession number GSE167097.

Mouse experiment: Animal care and procedures were conducted in accordance with institutional guidelines approved by the Institutional Animal Care and Use Committee (IACUC). The C57BL/KaLwRij mice (Harlan laboratories) were injected with 5TGM1-luc cells via tail vein. After the first day of engraftment, the mice were randomized and either treated with CCF642-34 per oral gavage (20 mg/kg) or with control vehicle (10% 2-hydroxy-propyl-β-cyclodextrin) three times a week for 8 weeks.

Myeloma patient database: In patients with plasma cell disorders who consented to Total Cancer Care^®^ Moffitt Cancer Center research sample protocol# 14690, IRB# Pro 00014441, CD138 magnetic bead purified bone marrow cells obtained during routine clinical bone marrow exams were submitted to RNA sequencing and results were annotated with key clinical variables such as plasma cell disorder type, number of prior therapies, survival, and refractory state to individual prior therapies via M2Gen/ORIEN. For the purpose of this manuscript all 689 MM patients from this registry were analyzed for PDIA1 expression and tertiles were subjected to Kaplan–Meier estimates for survival.

## 3. Results

### 3.1. Expression of PDIA1 Inversely Correlates with Survival in Patients with Relapsed or Refractory Myeloma

To understand if the PDIA1 expression correlates with survival we analyzed RNA-seq data of CD138-enriched bone marrow cells from MM patients. Among 690 MM patients seen at Moffitt Cancer Center and Research Institute, expression of PDIA1 as assessed by RNA sequencing separated patients into tertiles with significantly (*p* = 0.00012) inferior survival in the two higher tertiles (Figure 1A). When patients parsed by clinical groups as newly diagnosed myeloma (NDMM), early relapse (ERMM, 1–3 prior lines of therapy), or late relapse (LRMM > 3 prior lines of therapy) were analyzed separately, high PDIA1 expression conferred inferior survival in ERMM and LRMM but not NDMM (Figure 1B–D). This observation suggests that PDIA1 expression may confer adaptive resistance to available treatments. Accordingly, targeting PDIA1 may prove valuable not only because it blocks a very proximal step in the UPR, but, in addition, it may exploit a vulnerability of the resistance phenotype.

### 3.2. Development of a Potent PDIA1 Specific Inhibitor

The PDI inhibitor CCF642 was highly potent; however, it was limited for clinical development due to its insolubility and lack of bioavailability. To improve its solubility, potency, and selectivity, we used the CCF642 binding space in the catalytic site for in silico modeling and mapped the binding site to the helix-turn-helix motif composed of the WCGHCK binding site in the aa’ and bb’ domains of the PDIA1 active site (Figure 2A,B). The ligand plot analysis of the docked structure of CCF642 with PDIA1 suggested that we could engage specific pi stacking interaction with tryptophan W^396^ in the catalytic site with appropriate modifications of the *p*-methoxy group of the phenyl ring. We chose to accomplish this by attaching a 3-carbon flexible linker with a terminal primary amino group to the corresponding phenol followed by coupling to desired L-amino acids (Figure 2A; Appendix A). The N-BOC penultimate intermediates were purified using flash silica gel chromatography and then deprotected using 4N HCl in dioxane to generate the corresponding hydrochloride (HCL) salts. The compounds for testing were fully characterized by ^1^H, ^13^C NMR and high-resolution mass spectroscopy. Several of these new amino acid derivatives had increased solubility in aqueous buffer (confirmed in a turbidity assay (Appendix A)) as well as improved enzyme selectivity in comparison to parent compound CCF642. These results were consistent with their ClogP and LogS values (Appendix A) calculated by Rekker’s fragment system approach in ChemDraw [28,29]. The three different analogues containing tryptophan (CCF642-34), phenylalanine (CCF642-37), or tyrosine (CCF642-38) showed the most specific and up to 10-fold enhanced inhibition of PDIA1 as calculated using two independent substrates in cell free assays. The inactivation constant, K_inact_ of CCF642-34 for PDIA1 was found to be 88 ± 2.8 nM and 100 ± 8.5 nM in di-EGGS and insulin reduction assays, respectively (Figure 2C–F and Appendix A). The tryptophan analogue CCF642-34 was the most potent PDIA1 inhibitor in different assays among all derivatives of CCF642. Several of the amino acid substitutions, most notably histidine and alanine, were less effective as reflected in the ratio of k_2_/K_i_ (Appendix A), a marker for potency and selectivity [27]. Because CCF642-34 was the most potent analogue in cell free PDI assays (Figure 2E, F and Appendix A) and, accordingly, also the most potent in restricting the growth of MM1.S cells (Appendix A), we selected CCF642-34 as a lead compound for further analysis.

To identify binding partners of CCF642 and CCF642-34, which covalently bind to lysine in the PDIA1 active site [19], we synthesized biotinylated analogues, CCF642-Biotin and CCF642-34-biotin (Figure 2G; Appendix A). MM1.S cells were treated with the biotinylated derivatives and the cell lysate probed with streptavidin. As reported previously [19], CCF642 has off-target bindings in addition to its binding to PDIA1, however, its analogue CCF642-34 showed remarkable selectivity for PDIA1 (Figure 2G). When cells were treated for 6 h with CCF642-34-biotin followed by Western blot analysis using either PDIA1 or streptavidin-HRP antibodies, it was demonstrated that PDIA1 was the only specific target that appeared in our analysis (Figure 2G). CCF642-34-biotin showed two prominent bands when probed with streptavidin (Figure 2G). To confirm the identity of these two protein bands, first we probed them with anti-PDIA1 antibodies that showed the only one specific PDIA1 band corresponding to molecular weight 57 kDA, the lower band was not reactive with the PDIA1 antibody used in this assay. To test the identity of the lower band, we performed a streptavidin pull down followed by mass spectral analysis of all proteins. The lower and the upper bands were both PDIA1, as confirmed by the identity of the peptides in LCMS (Appendix A). We were able to map 80.11% and 84.80% peptides of the PDIA1 protein, upper and lower bands, respectively (Figure 2G). The PDIA1 has two active sites with identical structural arrangements around CGHCK motif, and the lower band’s mass spectral analysis confirmed its identity as the full-length protein, therefore, we concluded that it consisted of the breakdown products of ab and a’b’ fragments of PDIA1 that were not recognized by the antibody used in Western blot assays (Figure 2G; Appendix A). No other protein with significant scoring in LCMS/MS was enriched in a streptavidin pull down of MM1.S cells treated with CCF642-34-biotin, while several other proteins were pulled down with CCF642 as reported previously [19].

### 3.3. CCF642-34 Inhibits the Growth of Myeloma Cells without Any Significant Effects on Normal Bone Marrow-Derived CD34^+^ Cells

To support our findings that CCF642 analogues’ superior pharmacologic property and PDIA1 selectivity retains their ability to restrict the growth of the MM cells without inducing any adverse effect on normal bone marrow, we determined the LD50 of CCF642-34 against MM cells and found that, during in vitro cell culture, CCF642-34 demonstrated nearly 2-fold higher potency compared to CCF642. The LD50 of CCF642-34 on MM1.S was 118 ± 21 nM compared to 217 ± 19 nM for CCF642 (Figure 3A). The PDIA1 inhibitor CCF642-34 was also tested on additional multiple myeloma cell lines with different levels of PDIA1; KMS-12-PE, RPMI 8226, and U266, and the LD50 was 165 ± 8, 292 ± 11, and 371 ± 26 nanomolar, respectively (Appendix A).

Consistent with previous reports [21,30], we observed that BTZ-resistant MM1.S maintained its sensitivity to PDIA1 inhibition by CCF642-34. The resistant MM1.S cells had an LD50 of 60 ± 11 nM compared to 118 ± 21 nM for parental cells. (Figure 3B,E, Appendix A). Interestingly, BTZ-resistant MM1.S cells were ~2-fold more sensitive to PDIA1 inhibition compared to parental BTZ-naïve MM1.S cells. This effect in BTZ-resistant MM1.S cells may be in part be due to increased dependence of resistant MM1.S cells on PDIA1, reflected in adaptive increase of PDIA1 (Figure 3A,C,D).

In addition, we also examined if PDIA1 inhibition by CCF642-34 was synergistic or antagonistic to BTZ by Chou and Talalay assay [31]. Combined treatment of MM1.S-luc with CCF642-34 and BTZ demonstrated a clear synergy in the low dose range, which disappeared with an increasing concentration of either drug due to pronounced cell death. For the lower dose range, which affected 70% of myeloma cells or less, a synergistic combination index (CI) below 1 was observed for PDIA1 inhibition combined with BTZ in treatment-naïve MM1.S cells (Figure 3F and Appendix A). This synergy is likely explained by an increase in misfolded proteins upon inhibition of disulfide bond formation, leading to the greater dependence on proteasome to resolve ER stress [32].

Interestingly, CCF642-34 is 20-fold more potent in restricting the colony-forming abilities of MM cells, RPMI-8226, compared to its effect on the clonogenic potential of CD34^+^ HSPCs derived from healthy bone marrow, supporting PDIA1 as a target with favorable therapeutic index in multiple myeloma (Figure 3G–H, Appendix A).

### 3.4. CCF642 Analogues Induce Acute ER Stress Response Followed by Apoptosis in MM1.S Cells

To understand the mechanism of cell death induced by CCF642 analogues, we investigated the ER response and apoptosis. MM1.S cells were exposed to CCF642-34, CCF642-37, and also to the less effective analogue CCF642-39 as a control. While CCF642-34 and CCF642-37 induced a robust ER stress response, as evident from the induction of spliced X-Box Binding Protein-1S (XBP-1S) and C/EBP homologous protein (CHOP). The treatment of cells with inactive analogue CCF642-39 and CCF642-34A (Appendix A) failed to induce acute ER stress response, consistent with their lack of PDIA1 inhibition (Figure 4A). The induction of ER stress response is acute and leads to irreversible pro-apoptotic signaling as demonstrated by the extensive cleavage of PARP and caspase 3 (Figure 4B and Appendix A). ER stress was observed after 15 min of exposure (expression of XBP-1S and IRE1a oligomerization) and lasted for several hours in the presence of the drug (Figure 4B). The ER stress induced by PDI inhibition is irreversible and induces programmed cell death reflected in PARP and caspase 3 cleavage that starts ~1 h post-treatment (Figure 4B, Appendix A). As expected with PDI inhibition and ER stress, treatment with CCF642-34 robustly increased reactive oxygen species (ROS) in myeloma cells, observable within 25 min and peaking between 2–3 h where 4–6-fold increase was detected by carboxy-H_2_DCFDA (Figure 4C). As a result of increased ROS, we observed upregulation of NRF2 pathway genes (Appendix A)

### 3.5. Greater Selectivity of CCF642-34 for PDIA1 Inhibition Translates into a Narrower Band Gene Expression Profile Than CCF642

To investigate whether CCF642-34 affects gene expression changes that can be differentiated from CCF642, we performed transcriptomic profiling of MM1.S cells treated with either of these two compounds at 3 µM for 6 h or vehicle control using whole exome mRNA sequencing. Volcano plots were used to visualize differential expression (2-fold change with *p* value less than 0.05). Treatment of MM1.S cells with CCF642-34 or CCF642 changed the expression of 362 and 568 genes, respectively, compared to vehicle control (Figure 5A,B). Among these differentially expressed genes, 87 downregulated and 142 upregulated genes were common to both compounds, including downregulation of cell division and mitotic cell cycle processes and upregulation of response to ER stress, unfolded protein response, and apoptotic gene sets (Figure 5D–H, Appendix A). CCF642-34 treatment resulted in the down- and upregulation of 156 and 206 genes, whereas CCF642 caused down- and upregulation of 257 and 311 genes, respectively (Figure 5C). Hierarchical clustering showed distinct gene expression profiles in 642-34- and 642-treated MM1.S cells (Appendix A) and a narrower spectrum of genes involved in response to ER stress and UPR that was affected in expression after CCF642-34 compared to CCF642 treatment in MM1.S cells (Figure 5D,E). Consistent with the acute ER stress, a further gene set enrichment analysis suggested upregulation of more than half of the ER-associated PERK and ATF6 target genes expression after 6 h of CCF642-34 treatment (Figure 5F, Appendix A). In addition, genes associated with ubiquitin catabolism were also upregulated, a sign of the induction of unresolvable acute ER stress response caused by the accumulation of unfolded proteins. For example, the positive early sensor of ER stress response gene TRIB3 (Tribbles homolog 3), a negative regulator of NFkB that induces TRAIL and TNF activation-associated cell death [33,34], was nearly 20-fold upregulated compared to controls (Appendix A).

### 3.6. CCF642-34 Is Pharmacologically Stable to Acid Exposure and Does Not Undergo Rapid Hepatic Metabolism

To evaluate pharmacological properties of CCF642-34 we tested the stability in acidic conditions (6 N HCl) and upon exposure to human liver microsomes [21]. After exposure to acid for 3 h, greater than 80% of CCF642-34 could be recovered and the half-life in human liver microsomes was greater than 5 h, suggesting it would remain intact during gastric passage and not undergo substantial first-pass elimination (Figure 6A, Appendix A). Incubation with HCl or human liver microsome did not compromise the specific activity of CCF642-34. We therefore concluded that CCF642-34 may be orally bioavailable and effective against MM in vivo.

### 3.7. CCF642-34 Prolonged Survival of Mice in the 5TGM1 Syngeneic Mouse Model of Myeloma

To evaluate whether CCF642-34 achieved anti-myeloma efficacy after oral administration in vivo we used the 5TGM1-luc/C57BL/KaLwRij syngeneic mouse model. Two million 5TGM1-luc myeloma cells were injected by tail vein, and a week later treatment started with vehicle (10% 2-hydroxy-propyl-β-cyclodextrin *w*/*v* in water) or CCF642-34 dissolved in vehicle given by oral gavage 3 times a week for 8 weeks. Mouse weight was monitored along with systemic symptoms of distress or disease. According to IACUC protocol guidelines, a drop of 20% in body weight, paraparesis, or behavioral signs of distress constituted experimental endpoints and mandated euthanasia. All vehicle control animals required euthanasia or expired by 52 days, while 3 out of 6 CCF642-34-treated mice lived beyond 180 days with no sign of disease (Figure 6B). The intensity of luminescence was not good enough for the detection of bioluminescence, which is a common issue in the imaging of B57 black 6 mice. Therefore, we primarily monitored the survival, and the disease burden was determined in the bone marrow cells at the time of sacrifice of a moribund mice. The survival data were significant according to the Mantel–Cox test (*p* = 0.0391). Treatment caused no obvious adverse events as assessed by weight and animal behavior (data not shown). Results confirmed oral bioavailability and in vivo efficacy of CCF642-34 against myeloma.

## 4. Discussion

High baseline ER stress with an unfolded protein response (UPR) operating at capacity to prevent cell death is the result of high protein synthesis and secretion rate in neoplastic plasma cells that face micro-environmental stressors, which further increase the misfolded protein load. Protein homeostasis is central to the survival of highly proliferative malignant cells in general and MM cells in particular, which explains the efficacy of proteasome inhibition in the treatment of MM. As an incurable disease for the overwhelming majority of patients, with resistance developing to proteasome inhibitors and other novel drugs including CD38 antibodies, the treatment-refractory state of myeloma portends short survival below 6 months [6] and represents an unmet medical need. We found that patients with relapsed or refractory disease who expressed higher levels of PDIA1, the bottleneck enzyme for folding secreted proteins that contain intramolecular disulfide bonds, have inferior survival. These observations suggested that targeting PDIA1 could be an effective treatment strategy. The PDIA1 inhibition not only targets the overburdened protein synthesis of myeloma, but may also help overcome the treatment-refractory state of proteasome inhibitors. Building on the small molecule scaffold that inactivates PDIA1 by covalent attachment to lysine adjacent to its active site [19], we developed a pharmacologically improved analogue with greater solubility, selectivity, potency, and oral bioavailability that may serve as a lead for clinical translation.

Previously, we reported CCF642, a candidate small molecule PDI inhibitor with sub-micromolar IC50 with excellent safety in vitro and in vivo [19]. However, poor solubility and bioavailability were major hurdles for its clinical translation. Using a structure-guided medicinal chemistry approach, we significantly improved the solubility and in vitro efficacy as determined by the PDI inactivation constant K_inact_. This improvement in the potency was also reflected in the selectivity of the compound for PDIA1. The whole cell approach we used for evaluating the selectivity of CCF642-34 demonstrated highly preferential binding to PDIA1 without substantial off-target binding at nearly 60-fold above therapeutic (LD50) doses. CCF642-34 avoided off-target binding of CCF642 and affected the expression of a lower number of genes than CCF642 (Figure 5). It proved more potent against myeloma cells than its parent, the less PDIA1-selective compound CCF642 (Figure 3), and similarly induced the acute ER stress response that overwhelmed the capacity of myeloma cells to maintain proteostasis, hence leading to cell death (Figure 4). Consistent with the accumulation of misfolded proteins caused by inhibition of the ER resident PDIA1 enzyme, CCF642-34 treatment induced all three arms of ER stress response pathways [35,36]. Noticeably, the cleavage of XBP1 generating XBP1s as a result of IRE1 activation was seen as early as 15 min after treatment, consistent with an increase of misfolded proteins in the ER. The PERK-induced ATF4-associated target gene along with ATF6 target genes were significantly upregulated in the treatment group compared to vehicle, further supporting that inhibition of PDIA1 by CCF642-34 occurs in cells and leads to increase in misfolded ER proteins that are sensed by PERK (Figure 5F).

The kinetics of proteostasis of the secreted proteins in normal cells are guided by slower demand and are largely error-free; however, the malignant plasma cells that operate at maximum capacity can decrease the folding yield or rate of folding. The ER resident chaperones in such high stress conditions are unable to prevent the generation of toxic unfolded species. Indeed, the misfolded proteins in the ER are observed in some disease states that are known to program cell death, consistent with observations in MM cells [35,36]. Either the ER increases its ability to handle misfolded proteins, or misfolded proteins are destroyed, or the cell goes to apoptosis [37]. Dysregulation of unfolded protein response (UPR) and ER-associated degradation (ERAD) are exploited as MM cells’ vulnerability by PDIA1 inhibition.

One of the most striking observations was that CCF642-34 was active against proteasome inhibitor-resistant cells. In our analysis of relapsed and refractory MM patients, either early (ERMM, 1–3 prior lines of treatment) or late (LRMM, >3 prior lines), we observed an upregulation of PDIA1 expression, suggesting that the gain of ER function may contribute to the refractory state that results in poor survival. Most patients are treated upfront with proteasome inhibitors, and in the relapsed and refractory setting most have been exposed to two proteasome inhibitors. CCF642-34 had potent activity against myeloma cells that were made resistant to the proteasome inhibitor bortezomib (BTZ) through constant exposure, and a synergistic effect with bortezomib was observed in combination studies (Figure 3). CCF642-34 was highly stable in human liver microsomes and upon exposure to acid. As expected, based on these characteristics it proved effective upon oral administration in a well-established syngeneic mouse model of myeloma [37]. Our results are consistent with recent reports that found BTZ-resistant myeloma cells maintain sensitivity toward PDI inhibition [21,30].

## 5. Conclusions

In summary, we used structure-guided medicinal chemistry to develop a small molecule PDIA1 inhibitor with favorable solubility, selectivity, and potency that demonstrates pre-clinical evidence for bone-marrow-sparing anti-myeloma activity and bioavailability, supporting further development for a potential clinical utility.

## 6. Patents

M.H., D.R.G., J.P.M., F.J.R., J.G.P. and B.K.J. are inventors on the patent application filed by the Cleveland Clinic on the PDIA1 inhibitors reported in this manuscript.

## Figures and Tables

**Figure 1 cancers-13-02649-f001:**
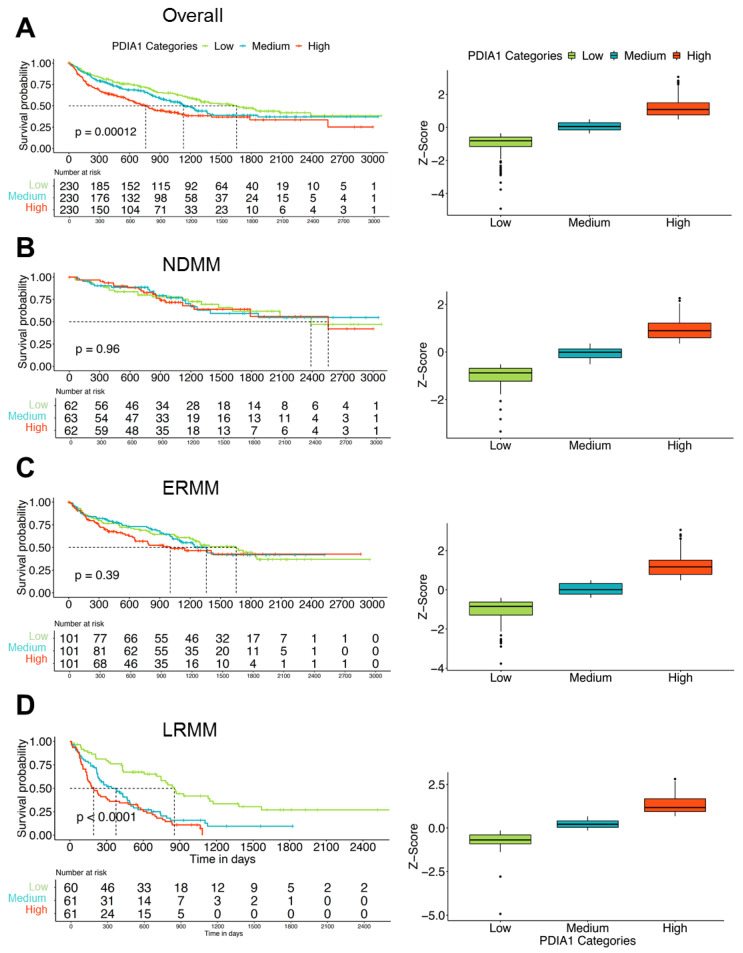
Expression of PDIA1 in myeloma associates with survival. CD138-purified myeloma cells from patients with multiple myeloma were subjected to RNA sequencing and survival was estimated based on Kaplan–Meier for low, medium, and high PDIA1 expression tertiles. The statistical significance for the survival curves was determined by Log-rank test. (**A**) High and medium expression resulted in inferior survival in the entire myeloma cohort. While (**B**) no significant effect on survival was seen in newly diagnosed myeloma, in (**C**) patients with early (1–3 prior lines of therapy) or (**D**) late relapse (>3 prior lines of therapy) medium and high PDIA1 expression tertile were associated with shorter survival. Abbreviations: NDMM—newly diagnosed multiple myeloma; ERMM—early relapse multiple myeloma; LRMM—late relapse multiple myeloma; os—overall survival.

**Figure 2 cancers-13-02649-f002:**
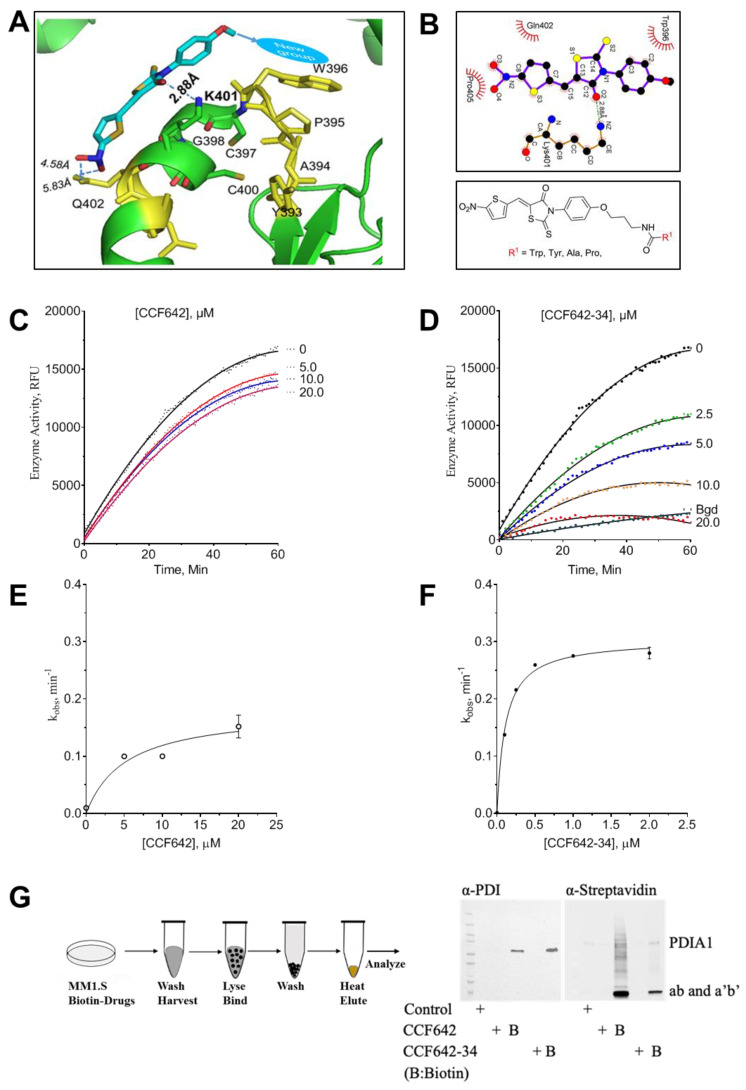
Model-based design improves potency and selectivity of protein disulfide isomerase A1 (PDIA1) inhibitors. (**A**) Ribbon diagram of CCF642-34 docked onto PDIA1. (**B**) Two-dimensional structure of modified CCF642 pharmacophore, where R represents amino acids tryptophan, tyrosine, phenylalanine, histidine, proline, or alanine. (**C**) PDIA1/PH4B activity assay was performed in the absence or presence of CCF642 or (**D**) CCF642-34. In time-dependent inhibition of di-E-GSSG, reduction was monitored for 1 h by the increase in fluorescence, and the relative fluorescence unit was plotted as a function of time. (**E**,**F**) The observed rate constant for inhibition, k_obs_, at each concentration determined from the slope of kinetic data presented in panel (**C**,**D**). The kobs values are re-plotted against inhibitor concentration and fitted to a hyperbolic equation, k_obs_ = k2[I]/(Ki + [I]), to obtain values for Ki and k2 in GraphPad Prism v8.0.2.The concentration of drug is indicated on each curve, for (**E**) 642 and (**F**) CCF642-34. (**G**) Target validation. Multiple myeloma cells (MM1.S) were treated with vehicle (DMSO) or B-CCF642-34 for 3 h and lysates were separated on SDS-PAGE gel followed by visualization by either anti-PDIA1 antibody or HRP-conjugated streptavidin. The bands’ identities are as labeled.

**Figure 3 cancers-13-02649-f003:**
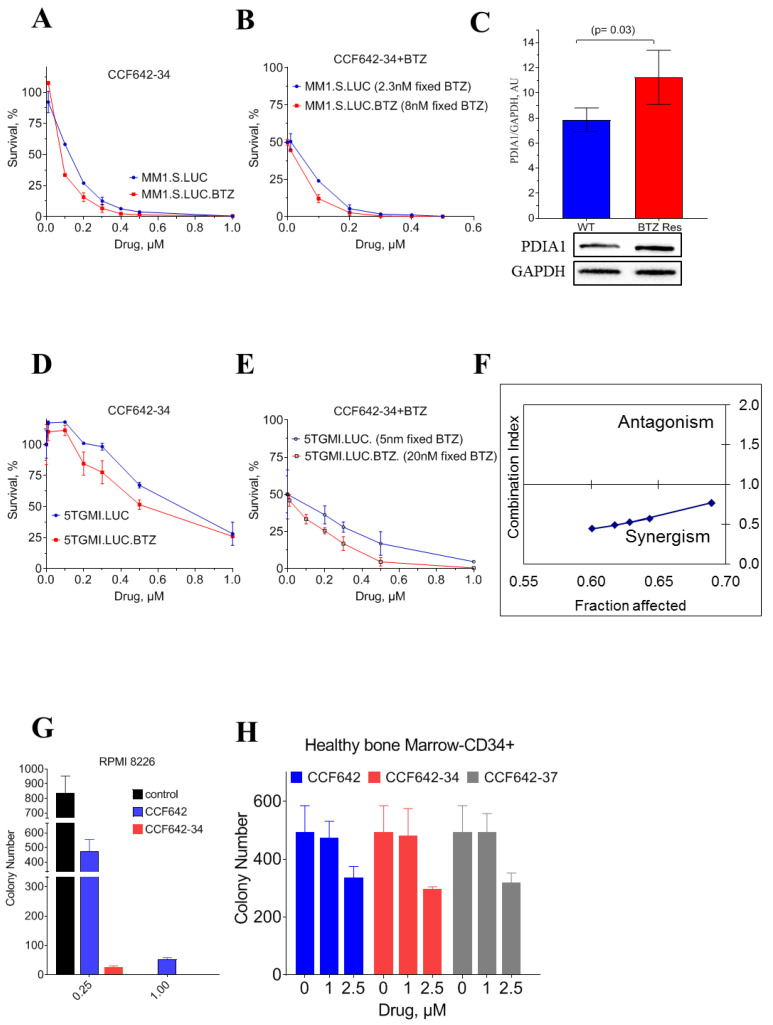
Selective cytotoxicity of PDIA1 inhibitor against multiple myeloma MM1.S cells. Cell viability and LD50 for inhibitors were measured in 96 well culture plates (2 × 10^4^ cells/well) after 72 h of treatment using CellTiter-Glo^®^ Luminescent Cell Viability Assay (Promega, Madison, WI, USA). (**A**) Cell survival assay with MM1.S.luc and BTZ-resistant MM1.S.luc cells. (**B**) Bortezomib was used at a fixed IC50 concentration of 2 nM for MM1.S.luc, 8 nM for MM1S.luc BTZ for combined drug toxicity analysis. (**C**) Comparison of PDIA1 protein levels in MM1.S.luc and MM1.S.luc BTZ-resistant cells. Band intensity was calculated with Image Lab Version 5.2.1. (**D**) Cell survival assay with 5TGM1.luc and (**E**) BTZ-resistant 5TGM1.luc cells. (**F**) MM1.S cell line was exposed to CCF642-34, and BTZ for 72 h for synergistic drug combination test according to Chou and Talalay method. If the fraction of cells affected remained less than 1, the two drugs were determined as synergistic. (**G**,**H**) Toxicity of CCF642 and its analogues, CCF642-34 and CCF642-37, against RPMI 8226 and CD34^+^ normal bone marrow cells from healthy individuals in a colony-forming assay. The number of colony-forming units were plotted for each treatment. The toxicity against normal bone marrow was estimated at ~20-fold over the drugs against multiple myeloma cells.

**Figure 4 cancers-13-02649-f004:**
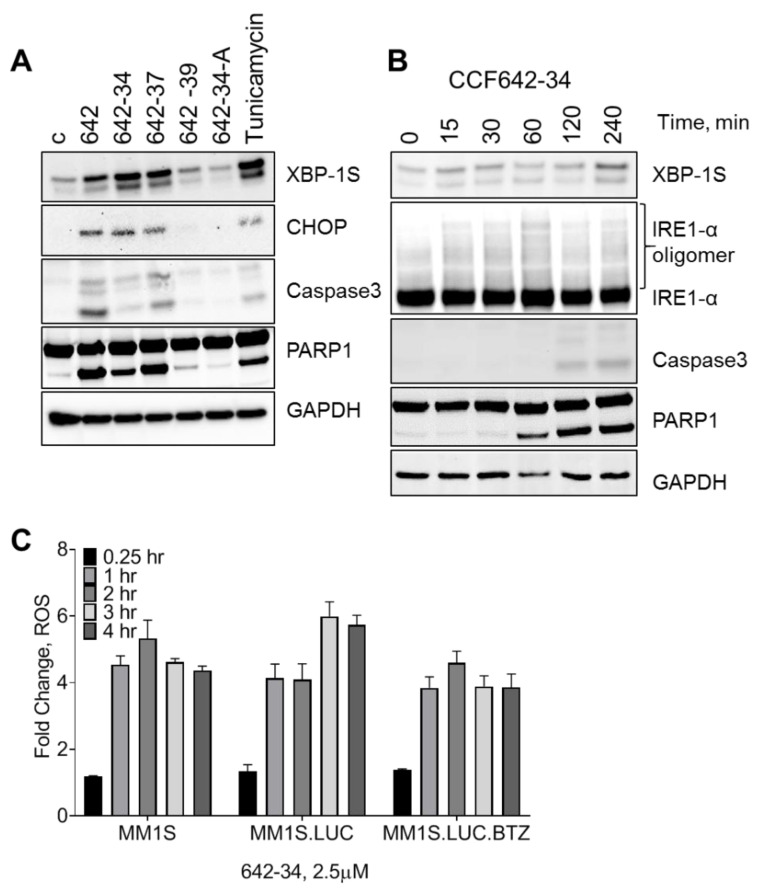
PDIA1 inhibition by CCF642 analogues induce acute endoplasmic reticulum (ER) stress response and lead to apoptosis. Multiple myeloma cells (MM1.S) were treated with 3 µM of CCF642, -34, -37, -39, -34-A, and Tunicamycin (as a control). The status of ER sensors (XBP-1S, IRE1α oligomerization, and induction of C/EBP homology protein (CHOP)) along with apoptosis markers (cleaved caspase 3 and PARP1) were monitored. (**A**) MM1.S cells were treated for 4 h with CCF642 and its indicated analogues. (**B**) MM1.S cells were treated with CCF642-34 in a time course. (**C**) MM1.S, MM1.S.luc, and BTZ-resistant MM1.S.luc cells were stained with the H2DCFDA ROS detection and then treated with 2.5 µM CCF642-34 up to 4 h.

**Figure 5 cancers-13-02649-f005:**
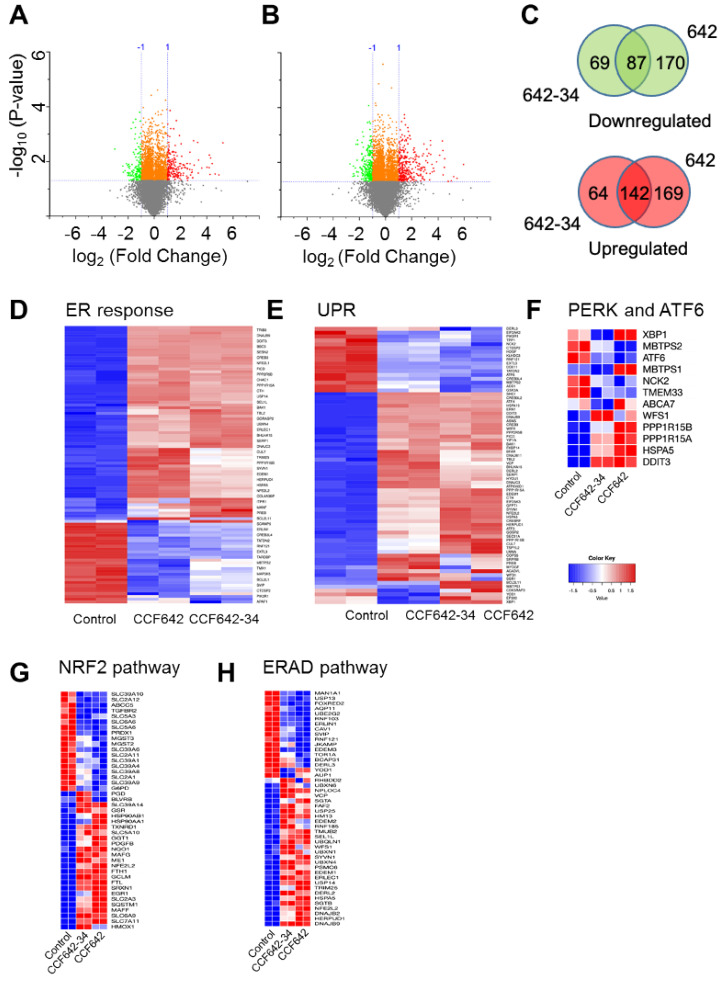
CCF642-34 is selective for PDIA1 inhibition-induced ER stress response pathway. MM1.S cells were treated with 3 µM of either CCF642 or CCF642-34 for 6 h, and gene expression analysis was performed by RNA sequencing. (**A**,**B**) Volcano plots showing CCF642-34 and CCF642 compared to DMSO control. The criteria for differential expression were at least a 2-fold change with *p* value less than 0.05. The analysis was performed in Originlab, Version 2019b (OriginLab Corporation, Northampton, MA, USA). (**C**) Differentially upregulated (red) and downregulated (green) genes between CCF642-34 and CCF642 were compared in Venn diagrams. The diagram was generated using a web-based server, http://bioinformatics.psb.ugent.be/webtools/Venn/ (accessed on 3 March 2020). (**D**,**E**) Hierarchical clusterings with heat map of MM1.S cells treated with vehicle CCF642-34 or CCF642 are shown for response to endoplasmic reticulum stress and unfolded protein response gene sets. (**F**–**H**) Heat maps of PERK and ATF6 target genes, Nrf2, and ER-associated degradation (ERAD) pathways are compared between control and inhibitor-treated MM1.S cells. Data analysis was performed using a web-based server http://bioinformatics.sdstate.edu/idep/ and https://software.broadinstitute.org/morpheus/ (accessed on 5 February 2021).

**Figure 6 cancers-13-02649-f006:**
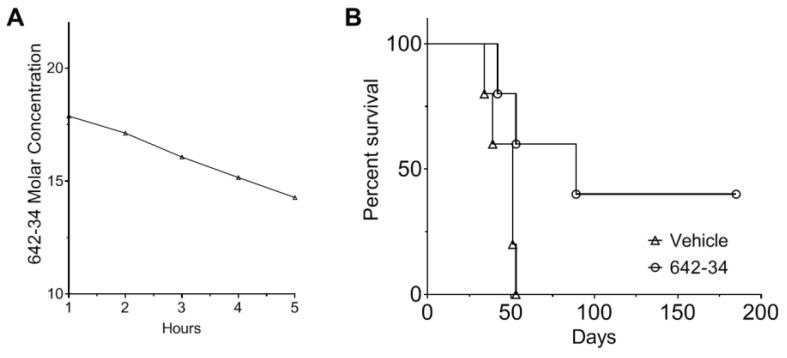
Stability of CCF642-34 and restriction of multiple myeloma in a syngeneic mouse model by PDIA1-inhibitor CCF642-34. (**A**) The stability of CCF642-34 was measured against oxidative metabolism by human liver microsomes. CCF642-34 (20 µM) was incubated with 0.25 mg/mL of human liver microsomes for 5 h. The residual compound at indicated time points was measured by HPLC (Agilent 1260 Infinity II) interfaced with reverse phase C18 column using 280 nm and 245 nm detection wavelength. The standard curve of known concentrations of CCF642-34 was obtained from the area under the peak at two wavelengths, and the remaining CCF642-34 was estimated. Data are representative of two independent experiments. (**B**) 5TGM1-luc/C57BL/KaLwRij mouse models of myeloma (3 males, 3 females per treatment group) were engrafted with 2 × 10^6^ 5TGM1-luc cells via tail vein injection and treated 3 times a week for 8 weeks per oral gavage with 20 mg/kg of CCF642-34 dissolved in 10% 2-hydroxy-propyl-β-cyclodextrin. The survival of each group was monitored, and Kaplan–Meier survival analysis was performed. All control animals required euthanasia before 52 days (due to paraparesis, weight loss, poor general condition), while 3 out of 6 mice treated with CCF642-34 lived beyond 6 months with no sign of disease.

**Table 1 cancers-13-02649-t001:** Cell line and culture condition.

Cell Line	Organism	Disease	Source	Media
MM1.S.luc	human	IgA lambda myeloma	ATCC	RPMI-1640 *
MM1.S.luc Btz^R^	human	Ig A lambda myeloma	CCF Lab	
RPMI-8226	human	Plasmacytoma; myeloma	ATCC	
5TGM1-luc	murine	Plasmacytoma; myeloma	Gift from Dr. Oyajobi [13]	IMDM *

* All cell culture media contained 10% fetal bovine serum (FBS, Bio-Techne Cat #S11150) and 1% P/S.

## Data Availability

Any specific material request can be made to the corresponding authors. All unique cell lines generated and used in this manuscript are available upon request with applicable institutional guidelines. The mass spectrometry proteomics data for PDIA1 characterization were deposited in the ProteomeXchange consortium via the PRIDE59 partner repository and the RNA-seq data were submitted to the Gene Expression Omnibus (GEO) repository at the National Center for Biotechnology Information (NCBI) archives with accession number GSE167097.

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
