# Peer review of "Therapeutic Targeting of Protein Disulfide Isomerase PDIA1 in Multiple Myeloma"

_cancers, 2021, doi:10.3390/cancers13112649_

Round 1

Reviewer 1 Report

This study evaluates the pre-clinical efficacy of a PDIA1 inhibitor in MM. The authors have previously developed PDIA1 inhibitors and evaluated their efficacy in MM and this study extends on those findings with an improved compound. PDIA1 presents as an interesting therapeutic target, particularly for a highly secretory malignancy such as myeloma. Furthermore the inhibitor demonstrates efficacy in bortezomib resistant cell lines. Overll this  study is of interest, however an number of issues must be addressed before publication.

  1. line 30 - should read high secretory burden of immunoglobulins 'and' cytokines.
  2. Include details for ROS detection and transcriptomic studies in methods. Has the transcriptomic data been made publicly available?
  3. In the legend for Figure 1, please state how the significance was determined for the survival curves, e.g. using the Log-rank test?
  4. Line 325-327 states that CCF642-34 was tested on a range of MM cell lines with different expression levels of PDIA1. This data should be included at least in supplementary (Western blotting and dose response curves). Is there any correlation with PDIA1 expression and sensitivity to CCF642-34?
  5. Considering the LD50 value of MM.1S cells is given as 118 nM, it is not clear why a dose of 3 uM was chosen for looking at the ER stress response by Western and for transcriptomic studies?
  6. It is stated that the bortezomib resistnat cell line MM.1S.LUC.BTZ are resistant to up to 8 nM bortezomib, however in the bortezomib and CCF642-34 combination curves it would appear that 5nM bortezomib alone reduces viability of this cell line to 50% - could you explain why?
  7. For figure legend 3, please explain the figure parts in order and indicate what time post-treatment that the viability was measured.
  8. Since luc-tagged cells were used for in vivo studies, was bioluminescent imaging performed to monitor disease burden? Addition of this data would greatly strengthen the in vivo data.

Author Response

Dear Reviewer, 

We sincerely appreciate your constructive criticism. Your suggestions have been well taken and your questions have been answered. Please see attached PDF file for our responses. 

Sincerely, 

Metis Hasipek

Reviewer 2 Report

The authors present novel preclinical data of the potent orally 35 bioavailable small molecule PDIA1 inhibitor CCF642-34. The methods described are scientifically sound and the results can serve as a basis for further development of this new drug candidate. Only a few minor point to consider:

1) Are there any data regarding the efficacy of this new drug against RRMM after previous therapy with IMiD or anti-CD38 mab? Currently, there are multiple treatment options for patients with RRMM and thus the key question is finding a novel treatment against these heavily pretreated patients.

2) Please elaborate more in the discussion regarding the potential of off-target effects and toxicities. Please also describe the UPR function in normal cells and the potential impact of its inhibition.

3) Please do not repeat three times in the manuscript the phrase that UPR is the "Achilles heel" of MM

Author Response

(The authors gave the same response as above.)

Round 2

Reviewer 1 Report

The authors have addressed the comments appropriately